# An analysis of frailty and multimorbidity in 20,566 UK Biobank participants with type 2 diabetes

Peter Hanlon [1✉], Bhautesh D. Jani[1], Elaine Butterly [1], Barbara Nicholl[1], Jim Lewsey [1], David A. McAllister[1,2] & Frances S. Mair[1,2]

## Abstract

**Background** Frailty and multimorbidity are common in type 2 diabetes (T2D), including people <65 years. Guidelines recommend adjustment of treatment targets in people with frailty or multimorbidity. It is unclear how recommendations to adjust treatment targets in people with frailty or multimorbidity should be applied to different ages. We assess implications of frailty/multimorbidity in middle/older-aged people with T2D.

**Methods** We analysed UK Biobank participants ($n = 20{,}566$) with T2D aged 40–72 years comparing two frailty measures (Fried frailty phenotype and Rockwood frailty index) and two multimorbidity measures (Charlson Comorbidity index and count of long-term conditions (LTCs)). Outcomes were mortality, Major Adverse Cardiovascular Event (MACE), hospitalization with hypoglycaemia or fall/fracture.

**Results** Here we show that choice of measure influences the population identified: 42% of participants are frail or multimorbid by at least one measure; 2.2% by all four measures. Each measure is associated with mortality, MACE, hypoglycaemia, and fall or fracture. The absolute 5-year mortality risk is higher in older versus younger participants with a given level of frailty (e.g. 1.9%, and 9.9% in men aged 45 and 65, respectively, using frailty phenotype) or multimorbidity (e.g. 1.3%, and 7.8% in men with 4 LTCs aged 45 and 65, respectively). Using frailty phenotype, the relationship between higher HbA1c and mortality is stronger in frail compared with pre-frail or robust participants.

**Conclusions** Assessment of frailty/multimorbidity should be embedded within routine management of middle-aged and older people with T2D. Method of identification as well as features such as age impact baseline risk and should influence clinical decisions (e.g. glycaemic control).

### Plain language summary

People living with type 2 diabetes often have multiple other long-term conditions (multimorbidity) or increased vulnerability to aging-related declines in health (frailty). These states are common in older people, however their prevalence and impact in people aged under 65 years are less clear. This study uses data from UK Biobank, a large group of people aged 40–72 years old, to study the impact of frailty and multimorbidity in relatively younger people with type 2 diabetes. We found that both frailty and multimorbidity were common in people with type 2 diabetes, even at relatively younger ages. People living with frailty or multimorbidity were at greater risk of mortality, heart attacks or strokes, falls or fractures, and of being hospitalized with low blood sugar. Assessing frailty and multimorbidity may help to identify people requiring individualized clinical management and assessment of risk.

[1] Institute of Health and Wellbeing, University of Glasgow, Glasgow, UK. [2] These authors jointly supervised: David A. McAllister, Frances S. Mair.
✉email: Peter.hanlon@glasgow.ac.uk

Type 2 diabetes mellitus is increasingly common, with prevalence rising with age[1]. Aging populations across the world present a growing challenge for the management of diabetes[2]. Type 2 diabetes is associated with states linked to the aging process[3,4], such as frailty and multimorbidity[3,5]. While the majority of people living with frailty are aged over 65 years (with prevalence rising steeply above this age), both frailty and multimorbidity are also often present in 'middle-aged' people with type 2 diabetes[5,6]. However, the clinical implications of these concepts in younger people are less well understood.

Frailty and multimorbidity are related but distinct concepts[7]. Neither has a universally accepted definition[8,9]. Frailty describes a dynamic state of increased vulnerability to decompensation in response to physiological stress, characterized by the reduced physiological reserve[10]. The two most common definitions are the frailty phenotype[11] and the frailty index[12]. Multimorbidity refers to the presence of two or more long-term conditions (LTCs) within an individual[8]. Multimorbidity is often quantified using a count of conditions, sometimes weighted depending on nature or severity[8]. Counts vary, however, in the number and type of conditions included. In both frailty and multimorbidity, the choice of definition dictates which individuals are identified as frail or multimorbid, and the degree of overlap between definitions is variable.

Guidelines for type 2 diabetes are beginning to recognize the importance of identifying frailty in older people with type 2 diabetes and tailoring management accordingly[13,14]. Specifically, targets for HbA1c should be relaxed in people with frailty or multimorbidity[13]. The rationale for less stringent targets in this context includes shorter life expectancy, as well as increased vulnerability to serious adverse effects of hypoglycaemia[15].

However, guidelines do not offer tailored guidance as to what degree of multimorbidity or frailty may alter the balance of risks and benefits in favour of more relaxed glycaemic targets, or indeed what conditions should be included in an assessment of multimorbidity. Importantly, it is not clear if the recommendations around glycaemic targets hold for younger people with type 2 diabetes who meet the definition of frailty or have multimorbidity[5,6].

To address this evidence gap, this study aims, in UK Biobank participants aged 40–72 with type 2 diabetes, to: (i) describe the prevalence of both multimorbidity and frailty using a range of possible definitions; (ii) assess the overlap between each definition of multimorbidity and frailty; (iii) compare the association between multimorbidity/frailty and adverse outcomes; and (iv) quantify the association between glycaemia (HbA1c) and adverse outcomes in people with and without frailty/multimorbidity.

We show that both frailty and multimorbidity are common in middle-aged people with type 2 diabetes, although different measures of each construct identify different individuals. We also show that, regardless of the measure used, frailty and multimorbidity are both associated with increased risk of mortality, major adverse cardiovascular events, falls or fractures, and hypoglycaemia. However, at a given level of frailty or multimorbidity, the absolute risk of each of these outcomes is higher among older people.

## Methods

**Study population**. This is an analysis of UK Biobank participants with type 2 diabetes. Participants were recruited between 2006–2010 by postal invitation and attended one of 22 assessment centers in England, Scotland, or Wales where they completed a touchscreen questionnaire, a nurse interview, had physical measurements, and provided blood samples. Participants also consented to data linkage to healthcare records including mortality and hospital episode statistics. Participants with type 2 diabetes were identified according to the validated algorithm developed by Eastwood et al.[16]. The UK Biobank has full ethical approval from the NHS National Research Ethics Service (16/NW/0274). All participants gave informed consent for participation in UK Biobank. Permission to access and analyse UK Biobank data was approved under UK Biobank project 14151.

**Measures: multimorbidity**. For this analysis, we compared two measures of multimorbidity: the Charlson Comorbidity Index[17], and a numerical count of long-term conditions[18]. For each score we removed diabetes, as type 2 diabetes is the index condition for the analysis. We chose the Charlson Comorbidity Index as it was recommended in a recent systematic review as the best tool to assess the risk of mortality in younger populations[19]. We also included a numerical count of LTCs, as this is a commonly used alternative to a weighted score[8,20]. Conditions were identified from self-report or from ICD-10 codes from hospital admission prior to baseline (code lists are detailed in supplementary data files 1 and 2).

The simple count was based on a list of 42 long-term conditions originally developed in a large epidemiological study in Scotland and subsequently adapted for UK Biobank. Conditions were identified based on either self-report or on ICD-10 codes from linked hospital episode statistics. Participants were considered to have a condition at baseline if they either reported the condition at the assessment centre nurse interview, or if they had a hospital admission prior to the assessment centre date with a relevant ICD-10 diagnostic code (see supplementary data 2 for relevant ICD-10 codes and self-reported condition included in each definition). The total number of conditions at baseline was summed to give an overall count.

Conditions included in the Carlson Comorbidity Index were similarly identified from self-report or from ICD-10 codes from hospital admission prior to baseline. ICD-10 codes were taken from a previously validated algorithm for administrative data. Each condition was then weighted (ranging from 1–6) according to the algorithm and the weights were summed to give a total score.

**Measures: frailty**. We assessed two operational measures of frailty at baseline: the frailty phenotype[11] and the frailty index[12]. These have both been adapted for use in UK Biobank[6,21].

The frailty phenotype was based on five criteria (low hand-grip strength, slow walking speed, weight loss, self-reported exhaustion, and low physical activity) and categorized as robust (0 criteria), pre-frail (1–2 criteria), and frail (≥3 criteria)[6,11]. Definitions were adapted to UK Biobank baseline data from the original description where required. Weight loss was self-reported according to the question "Compared with one year ago, has your weight changed?" (yes, reduced = 1, other response = 0). Exhaustion was assess using the question "Over the past two weeks, how often have you felt tired or had little energy?" (more than half the days or nearly every day = 1, other = 0). Slow walking pace was self-reported as "How would you describe your usual walking pace?" (slow = 1, other = 0). Physical activity was self-reported according to UK Biobank physical activity questionnaire. We classified the responses into: none (no physical activity in the last 4 weeks), low (light DIY activity [eg, pruning, watering the lawn] only in the past 4 weeks), medium (heavy DIY activity [eg, weeding, lawn mowing, carpentry and digging], walking for pleasure, or other exercises in the past 4 weeks), and high (strenuous sports in the past 4 weeks). Participants reporting none or light activity with a frequency of once per week or less were coded as 'low physical activity'. Grip strength was assessed

using a Jamar J00105 hydraulic hand dynamometer. The highest valid reading was used to classify grip strength according to cut-offs described by Fried et al.[11].

The frailty index is an unweighted count of 'deficits' which (i) increase in prevalence with age; (ii) are associated with poor health; and (iii) are neither ubiquitous in the population nor too rare (i.e. <1% prevalence)[12,22]. Deficits include long-term conditions, symptoms, and functional limitations. We used the list of deficits developed by Williams et al. for UK Biobank (excluding diabetes)[21]. This is summarized in supplementary data 3. The frailty index is calculated by dividing the number of deficits present by the total number of possible deficits, giving a value between 0 and 1 (higher values indicating a greater degree of frailty). Where an individual had missing data for a deficit, this deficit was also excluded from the demoninator[22].

**Measures: covariates**. All covariates used in analyses were based on baseline assessment centre data. Age and sex were used as recorded. BMI was calculated based on measured weight and height. Smoking was categorized as never, previous, or current based on self-report. The frequency of self-reported alcohol intake was categorized as never/special occasions only; 1–3 times per month, 1–4 times per week, and daily or almost daily. Townsend scores were calculated from postcode areas based on previous census data to give an area-based measure of socioeconomic deprivation[23]. HbA1c was taken from baseline blood samples obtained by UK Biobank.

**Outcomes**. Outcomes were identified by linkage to national mortality records (Office for National Statistics) and Hospital Episode Statistics. Linkage was carried out by UK Biobank and made available to approved researchers. The median follow-up was 8 years. Outcomes were all-cause mortality, cardiovascular mortality (the underlying cause of death ICD-10 code beginning with "I"), cancer mortality (ICD-10 code beginning with "C"),

Major Adverse Cardiovascular Event (MACE; cardiovascular death, or hospitalisations coded as non-fatal myocardial infarction [I21] or stroke [I63–I64]), hospitalization with hypoglycaemia (E16.0, E16.1, E16.2), and hospitalization with fall or fracture (W0, W1, S02, S12, S22, S32, S42, S52, S62, S72, S82, S92, T05).

**Statistical analysis**. We plotted the distribution of each frailty and multimorbidity measure descriptively. We then summarized the relationship between each measure and baseline characteristics by dividing each measure into four quartiles.

To assess the overlap between the four measures of frailty or multimorbidity we took all participants with scores above the 75th centile for each score (or the 'frail' category for the frailty phenotype). We then constructed a Venn diagram of the overlap between people above the 75th centile (or "frail" by frailty phenotype) for each measure.

To assess the relationship between each measure and clinical outcomes we used parametric survival models. We used Weibull models as this distribution was found to fit the data well for each measure and other covariates (assessed by plotting log time against the log of the estimated cumulative hazard). Models were adjusted for age, sex, ethnicity, socioeconomic status, BMI, smoking, and alcohol frequency. We modelled nonlinear effects of the frailty index, Charlson index, multimorbidity count, and age using fractional polynomials. We also assessed interactions between each measure and age, and between age and sex, and included these in the model where they were significant (p-interaction < 0.05). We modelled time to the first event. Competing risks were accounted for by using cause-specific models (i.e. participants were censored at the first occurrence of the outcome of interest, end of follow-up, or death, whichever occurred first. In models for MACE, falls or hypoglycaemia, deaths of other causes were coded as '0').

After fitting each model we predicted the 5-year risk of the incident outcome. Predictions were calculated separately for

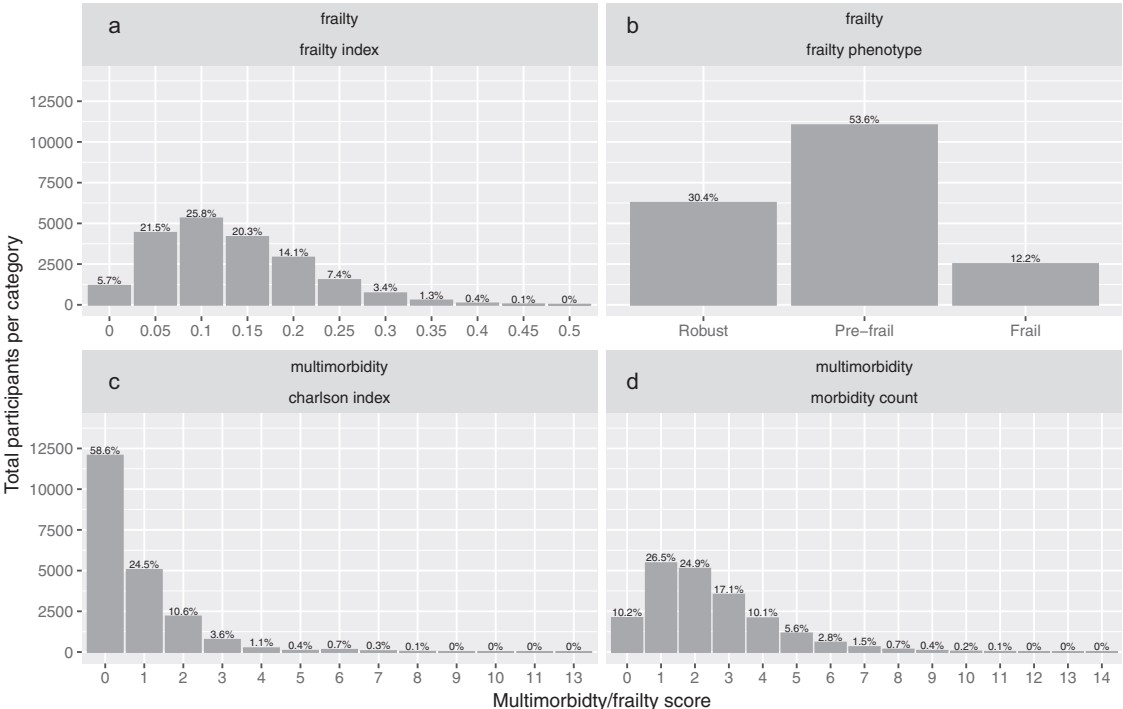

**Fig. 1 Distribution of frailty or multimorbidity.** This figure shows the distribution of each measure of frailty or multimorbidity (panel (**a**) frailty index, (**b**) frailty phenotype, (**c**) Charlson index, (**d**) long-term condition count). The height of the bar indicates the number of participants with percentages indicated above the bars.

males and females, holding age, BMI and socioeconomic status at the sample mean, smoking status as 'previous' and alcohol frequency as 1–4 times per week (the most numerous category).

Finally, to assess the impact of HbA1c on all-cause mortality at different levels of frailty or multimorbidity, we fitted Weibull models including HbA1c along with the covariates described above. Non-linear relationships between HbA1c and mortality were modelled using fractional polynomials. We included any significant interactions between HbA1c and frailty or multimorbidity. The predicted 5-year risk was calculated across all observed values of HbA1c, and at the 25, 50, 75, and 90th centiles of each frailty or multimorbidity definition (or at each category of the frailty phenotype). This allowed us to assess the relationship between HbA1c and mortality at different levels of multimorbidity or frailty.

All analyses were prespecified and reported according to the Strengthening Reporting of Observational Studies in Epidemiology (STROBE) statement (www.strobe-statement.org). Analyses were performed using R version 3.6.1. All syntax for deriving variables and for generating analysis will be returned to UK Biobank for record and will be available upon application to UK Biobank.

## Results

**Baseline characteristics**. A total of 20,566 UK Biobank participants were identified as having type 2 diabetes at baseline. The distribution of multimorbidity (defined by the Charlson Comorbidity Index and by a count of 42 long-term conditions) and frailty (defined by the frailty index and by the frailty phenotype) is shown in Fig. 1. Baseline characteristics and correlation between each of these measures are shown in the supplementary data files 4 and 5 and supplementary Fig. 1.

Most participants with type 2 diabetes were aged over 60 years (12,755, 62%). Only 1858 (9%) were aged under 50 years. The prevalence of frailty was broadly similar across age categories (e.g. frailty prevalence by frailty phenotype was 12.6% at age 40–50, 13.4% at age 50–60, and 11.5% at age 60–72; details in supplementary tables 1–4). The relationship with age varied between the individual components of the frailty phenotype. Low grip strength and slow walking speed increased in prevalence with increasing age, however low physical activity, self-reported exhaustion, and self-reported weight loss were more common in younger participants. However, the prevalence of multimorbidity with either measure rose with age (e.g. using the Charlson Comorbidity Index 7.7% scored ≥ 2 at age 40–50, 11.6% aged 50–60, and 20.6% aged 60–72).

The relationship between frailty and ethnicity differed depending on the frailty definition: compared to White participants, frailty is more common among Black and Asian participants when using the frailty phenotype definition, but less common when using the frailty index definition. Multimorbidity was less common among Black or Asian participants, compared to White. Both frailty and multimorbidity were strongly associated with socioeconomic deprivation by all definitions. Frailty phenotype (but not frailty index) were associated with slightly higher HbA1c. Participants with multimorbidity (using LTC count or Charlson) had lower mean HbA1c. However, in all cases, the differences were small (<2 mmol/mol) (supplementary data 4 and 5).

**Overlap between definitions**. There was relatively little overlap between the four measures of frailty or multimorbidity. Forty-two percent of participants were above the 75th percentile for at least one of the measures, but only 2.2% were identified by all 4 measures (Fig. 2). The correlation between measures is shown in Supplementary Fig. 1.

## Relationship between frailty or multimorbidity and outcomes

*Mortality.* Fig. 3 shows the adjusted 5-year mortality at different levels of frailty/multimorbidity. Higher degrees of frailty or multimorbidity were associated with greater all-cause mortality using each measure. The absolute mortality risk was higher at the extremes of the multimorbidity count and Charlson Index than for the frailty phenotype or frailty index, however there were also fewer participants with values at these extremes. Males had a higher mortality risk than females.

Age was a significant predictor of mortality risk, independent of frailty or multimorbidity. For example, using the frailty phenotype, the 5-year mortality for frailty was 1.9%, 4.4%, and 9.9% in men aged, 45, 55, and 65, respectively. For a multimorbidity count of 4, predicted 5-year mortality was 1.3%, 3.7%, and 7.8% in med aged 45, 55, and 65, respectively (supplementary data 6). There was no statistically significant interaction between age and any measure. Therefore, although the increase in relative risk associated with frailty or multimorbidity is similar across all ages studied, the absolute risk of mortality associated with any level of frailty or multimorbidity is higher at older ages.

These patterns were similar for cardiovascular mortality and for cancer mortality (supplementary data 6).

In post-hoc analyses, we assessed the relationship between the frailty phenotype and mortality within strata of multimorbidity (0, 1, 2, and 3 or more long-term conditions). At each level of multimorbidity, frailty was associated with an increased risk of mortality. Participants meeting the criteria for both frailty and multimorbidity had a greater risk of mortality than those meeting the criteria for frailty or multimorbidity alone.

*MACE, falls, and hypoglycaemia.* The estimated 5-year risk of incident hospital episode related to MACE, fall/fracture, or hypoglycaemia, are shown in Fig. 4. Each of these outcomes was associated with both frailty and multimorbidity. Female participants were at greater risk of falls/fractures. Males had a higher risk of MACE and hypoglycaemic hospitalization. As with mortality, the risk was highest at the extreme end of the distributions for the frailty index, multimorbidity count, and Charlson Index. Age was also a significant predictor of each outcome, with higher absolute risks among older participants at a given level of frailty or multimorbidity (supplementary data 6).

*HbA1c and all-cause mortality.* Fig. 5 presents the relationship between HbA1c and all-cause mortality at different levels of frailty or multimorbidity. Results were stratified according to centiles (25th, 50th, 75th, and 90th) of each measure, and categories of the frailty phenotype. The expected J-shaped relationship with mortality was observed throughout all levels apart from frail participants identified using the frailty phenotype, in whom the risk of mortality increased in a more linear fashion with increasing HbA1c. These analyses were repeated after stratifying by baseline use of drugs associated with hypoglycaemia (insulin and sulphonylureas). In participants who were frail according to the frailty phenotype, the steep rise in mortality risk with HbA1c was only observed in those not taking insulin or sulphonylureas at baseline. In participants taking these hypoglycaemic agents, the relationship between HbA1c and mortality was J-shaped for participants with frailty, as it was for pre-frail and robust participants (Supplementary Fig. 2).

## Discussion

Both frailty and multimorbidity were common at all ages in this cohort of 20,566 people with type 2 diabetes aged 40–72 years. Both the frailty phenotype and frailty index, as well as both

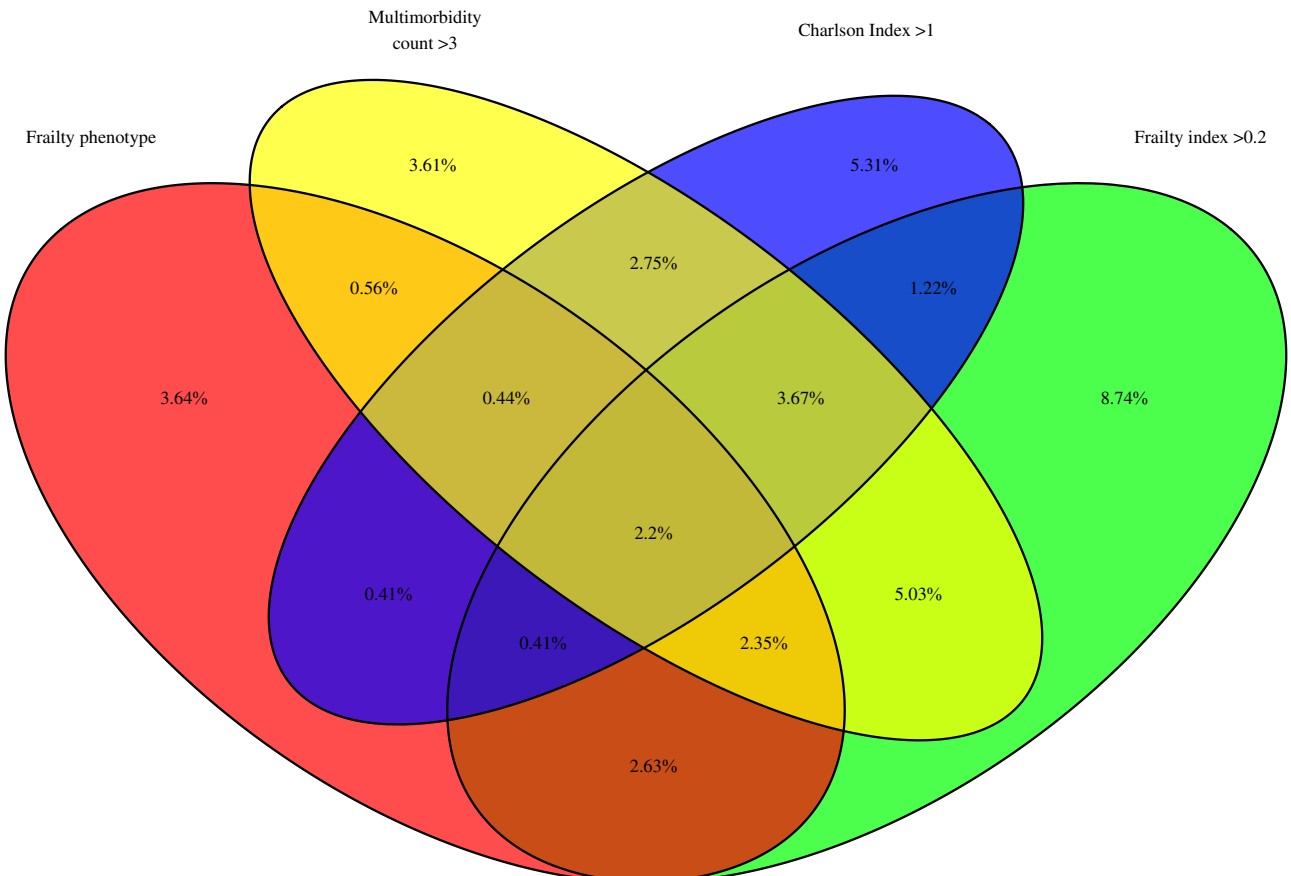

**Fig. 2 Venn diagram of overlap between frailty and multimorbidity measures.** This figure shows the overlap between each definition of frailty or multimorbidity. The percentage of participants identified by each combination of measures is shown by the percentages in each overlapping section. Note that 58% of participants were below the 75th centile for all definitions and are therefore not included in the Venn diagram.

weighted and unweighted measures of multimorbidity, identified people at greater risk of mortality as well as MACE and hospital admission resulting from falls, fractures, or hypoglycaemia. However, despite similarities in the risks associated with each measure, the participants who were identified as 'high risk' differed considerably between measures. Therefore, even in this relatively young population, frailty and multimorbidity identify people with type 2 diabetes at risk of a wide range of adverse outcomes, however relying on a single narrow construct may overlook others who may also be at higher risk.

Guidelines recommend higher glycaemic targets in people with frailty or substantial multimorbidity[13]. The higher mortality and risk of falls and hypoglycaemia that we observed in people with frailty or multimorbidity are consistent with the rationale for these higher targets: namely reduced life expectancy and greater risk of complications of hypoglycaemia[15]. However, our findings also demonstrate that the absolute risk of mortality in younger people with frailty or multimorbidity is considerably lower than in older people. Furthermore, the risk of all cause mortality among people identified as 'frail' using the frailty phenotype was highest among people with higher baseline HbA1c. This suggests that the implications of frailty or multimorbidity for clinical decision-making must rely on careful consideration of additional factors that influence baseline risk (including age) as well as individual patient preferences. This is important, as our findings suggest that frailty and multimorbidity are common among people with diabetes under the age of 65, however absolute risk of outcomes, and thus implications for clinical management, may differ at younger ages.

It is perhaps surprising that the prevalence of frailty, particularly using the frailty phenotype, did not increase with age. This was largely driven by a higher prevalence of low physical activity, self-reported exhaustion, and self-reported weight loss among younger participants. This could reflect a lack of specificity for these constructs in identifying frailty when applied to younger people, in whom characteristics such as exhaustion or weight loss may be biologically or phenotypically different from older people. It is important to note, also, that weight loss was not specified as unintentional in UK Biobank, limiting its specificity for indicating frailty. Finally, this relationship between frailty and age in this study could represent collider bias. For example, low physical activity in may have a causal relationship with the manifestation of type 2 diabetes at younger ages, but also with the identification of frailty, thus influencing the relationship between frailty and age when conditioning on a diagnosis of type 2 diabetes.

The finding that frailty and multimorbidity are common among middle-aged and older people with type 2 diabetes is consistent with previous studies[5,24]. So too is the increased risk of mortality and cardiovascular events[4,24]. The finding that the populations identified by each measure did not fully overlap is consistent with previous literature and is not surprising, given that these are distinct constructs underpinned by different models of frailty or multimorbidity[25]. Our findings add to this literature by demonstrating that even in this relatively young population, each measure identifies individuals at increased risk of adverse outcomes. Therefore, a narrow focus on a single measure may overlook others who are also at risk. Individualised person-centered care is likely to be appropriate and beneficial regardless

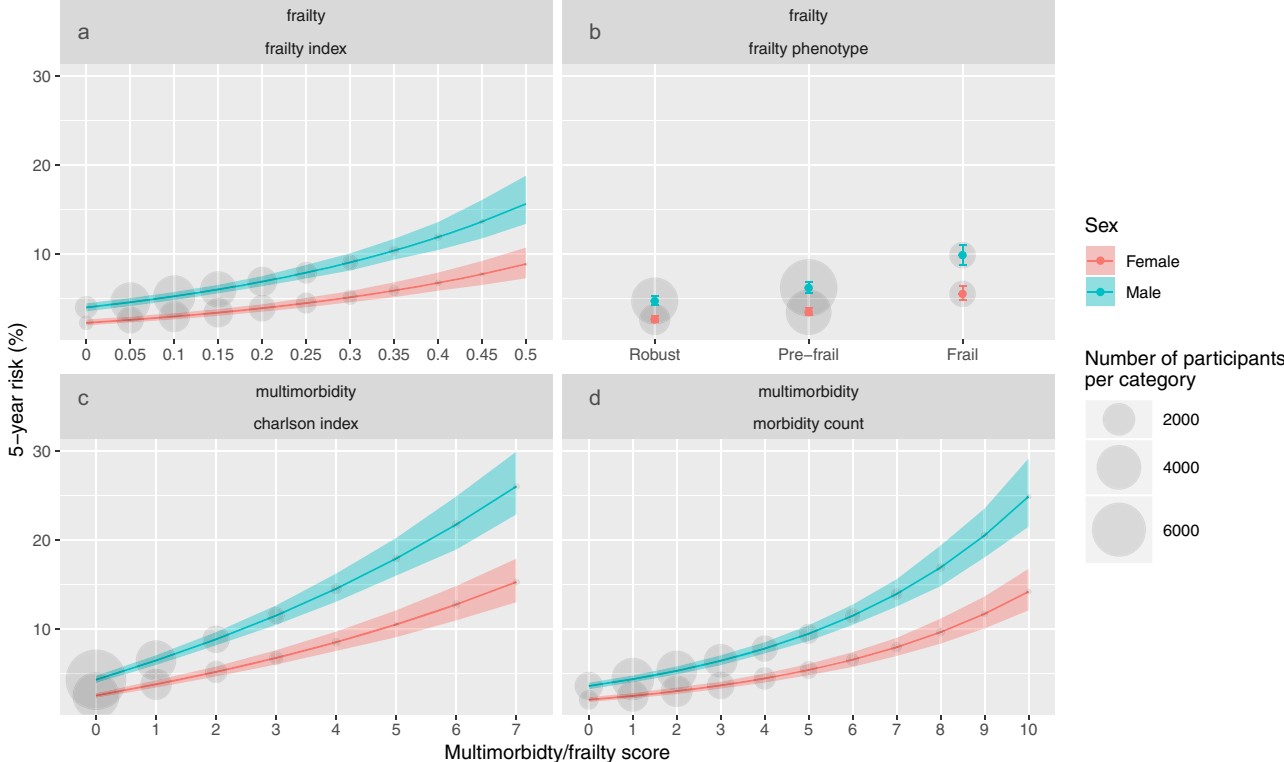

**Fig. 3 Relationship between frailty or multimorbidity and all-cause mortality.** This figure shows the predicted 5-year mortality rate for each measure of frailty or multimorbidity (panel (**a**) frailty index, (**b**) frailty phenotype, (**c**) Charlson index, (**d**) long-term condition count). Coloured lines or points indicate point estimates for predicted 5-year mortality. Men are shown in blue, and women in red. Shaded areas indicate 95% confidence intervals. Grey circles indicate the number of participants with each level of frailty or multimorbidity. Models are adjusted for age, sex, socioeconomic status, body mass index, smoking, and alcohol. Predicted 5-year mortality is based on age 60, socioeconomic status and body mass index held at the sample mean, previous smokers, and 1–4 times weekly alcohol intake.

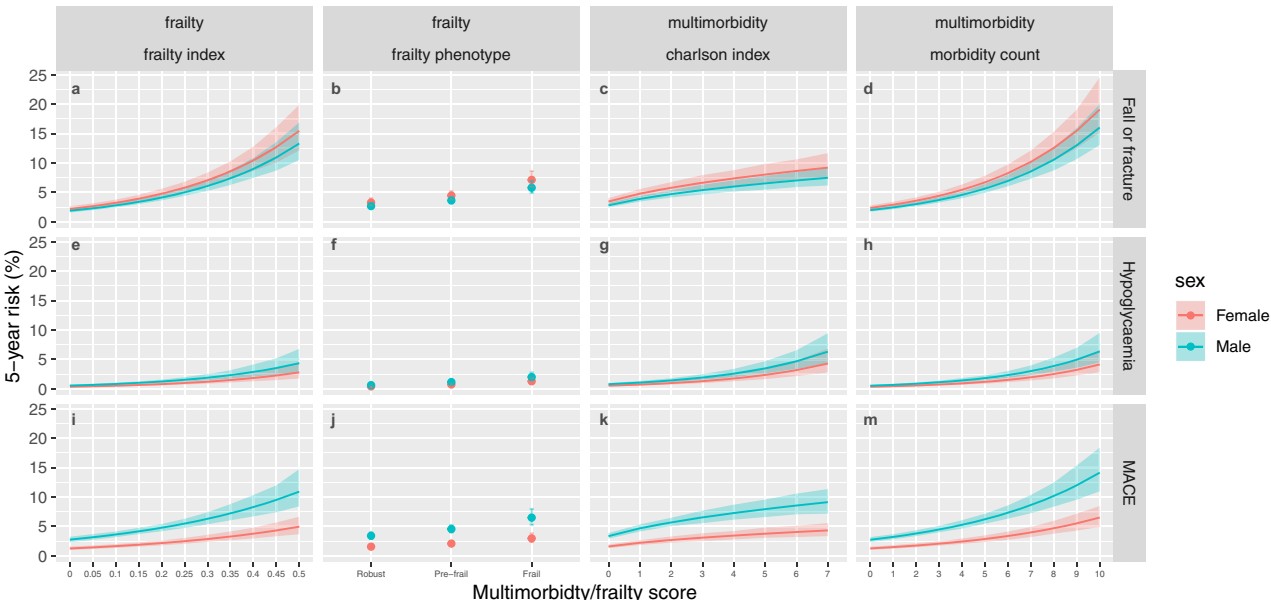

**Fig. 4 Relationship between frailty or multimorbidity and MACE, hypoglycaemia, and falls.** This figure shows the predicted 5-year rate of fall or fracture (panels **a**, **b**, **c**, and **d** showing the frailty index, frailty phenotype, Charlson index, and long-term condition count, respectively) hospitalization with hypoglycaemia (panels **e**, **f**, **g**, and **h** showing the frailty index, frailty phenotype, Charlson index, and long-term condition count, respectively) and MACE (panels **i**, **j**, **k** and **m** showing the frailty index, frailty phenotype, Charlson index, and long-term condition count, respectively). Coloured lines or points indicate point estimates for predicted 5-year mortality. Men are shown in blue, and women in red. Shaded areas indicate 95% confidence intervals. Models are adjusted for age, sex, socioeconomic status, body mass index, smoking, and alcohol. Predicted 5-year risk is based on age 60, socioeconomic status and body mass index held at the sample mean, previous smokers, and 1–4 times weekly alcohol intake.

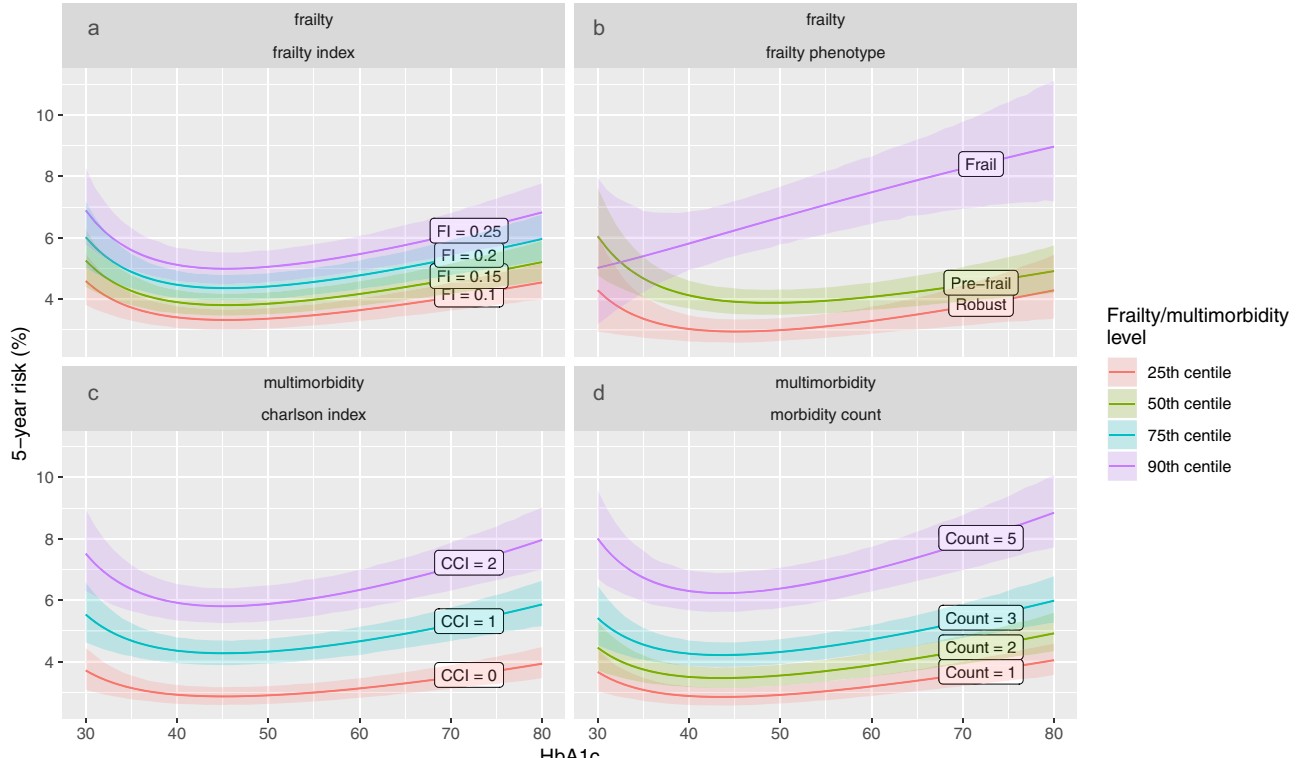

**Fig. 5 HbA1c and all-cause mortality.** This figure shows the relationship between HbA1c and predicted 5-year mortality at different levels of frailty or multimorbidity (panel (**a**) frailty index, (**b**) frailty phenotype, (**c**) Charlson index, (**d**) long-term condition count). Coloured lines or points indicate point estimates for predicted 5-year mortality. Colours indicate the level of frailty or multimorbidity according to centiles. Shaded areas indicate 95% confidence intervals. Models are adjusted for age, sex, socioeconomic status, body mass index, smoking, and alcohol. Predicted 5-year mortality is based on age 60, socioeconomic status, and body mass index held at the sample mean, previous smokers, and 1–4 times weekly. There was a significant interaction between the frailty phenotype and HbA1c. Interactions between frailty index, Charlson index, and LTC count were not significant.

of measure and further research, ideally using randomised trials, is required to understand if and how our approach should differ by how frailty or multimorbidity manifests.

The small magnitude of difference in HbA1c with frailty or multimorbidity identified is consistent with existing literature, the majority of which have shown no association with HbA1c[4,24,26]. This is perhaps surprising given guidelines for lower targets. Our findings may reflect the relatively young age of this cohort. However, others have observed hypoglycaemic medications are rarely discontinued in patients with frailty and low HbA1c, despite the risk of hypoglycaemia that this presents[27].

The relationship between HbA1c and mortality in people classified as 'frail' using the frailty phenotype is surprising, as we had expected the risks associated with lower HbA1c to be higher in people with frailty. Further analyses stratified by baseline use of hypoglycaemic agents suggest that low HbA1c in the context of insulin or sulphonylurea use (potentially reflecting over-treatment) is associated with increased mortality regardless of frailty status. Patients with low HbA1c may therefore benefit from deprescribing or dose reduction. The steeper rise in mortality with higher HbA1c and frailty was mostly driven by participants not taking these agents and may reflect a greater risk of sub-optimal glycaemic control in younger people living with frailty. This finding would need to be verified in other cohorts, and also explored further in older populations which represent the majority of people living with frailty.

Few studies have assessed the relationship between frailty and hypoglycaemia[24]. Several studies, mostly using the Charlson comorbidity index, have shown an increased risk of hypogly-caemia associated with multimorbidity[4]. Evidence linking frailty

with hypoglycaemia has been based on findings from trials such as ACCORD where patients over 80 years old had high rates of hypoglycaemia when randomised to the intervention arm[28], as well as the fact that older people appear most likely to be hos-pitalised with hypoglycaemic complications[29–31]. In both cases frailty has been hypothesized to explain the underlying vulner-ability. Our findings are concordant with this hypothesis and suggest that frailty may also confer some increased risk at younger ages.

Clinicians managing type 2 diabetes are likely to encounter high levels of frailty and multimorbidity, even among relatively young patient populations. Guideline recommendations for less stringent glycaemic targets in people living with frailty are in part predicated on limited life expectancy[13–15]. Our findings demon-strate that both frailty and age are important predictors of mor-tality risk, and while younger people with type 2 diabetes may meet the criteria for frailty, their absolute risk of mortality may be considerably less than an older person identified as frail. Fur-thermore, the choice of measure for frailty or multimorbidity substantially impacts which individuals are identified as 'high risk', with only partial overlap between definitions. These obser-vations, consistent with previous literature, are important in this context as diabetes guidelines do not currently give recommen-dations for how frailty in younger people should influence management (and in whom the assumptions around life expec-tancy underpinning recommendation for older people are unli-kely to hold) or how frailty and multimorbidity should be identified. Within populations identified as frail or multimorbid there is considerable heterogeneity in personal characteristics as well as variation in risk of adverse outcomes. This highlights the

importance of individualised decision-making for patients, taking into account patients' age and the measure used to assess frailty and multimorbidity, rather than blanket recommendations for 'frailty' or 'multimorbidity'. So, while a recent systematic review has suggested the need to embed screening for frailty within routine diabetes reviews[24], this work suggests that clinicians need to ensure care is tailored to the potential needs of people with frailty or multimorbidity taking into account of a wide range of factors. While frailty and multimorbidity do indicate gradients of risk, it may be that these are not the optimal tools to assess the appropriate targets for treatment in middle-aged people.

The strengths of this study include its large sample size with linkage to mortality and hospital event data. We also used a range of definitions of frailty and multimorbidity, which is an advantage as comparisons between studies are often limited by differences in the definitions used. Our focus on younger people than most previous frailty studies is relatively novel, as the implications of frailty in younger ages are not well understood. However, our findings may not be entirely transferable to older people (>70 years), in whom frailty is both more prevalent and may have a greater impact. UK Biobank was not specifically designed to assess frailty or aging, which limits our assessment of frailty. Specifically, some of the frailty phenotype components were adapted (e.g. weight loss was self-reported and not specifically unintentional) and the frailty index, while constructed according to standard guidelines, contains relatively few functional and sensory deficits.

Our analysis was limited by only having access to baseline measures of frailty and multimorbidity, as well as covariates such as HbA1c and body mass index. Both frailty and multimorbidity are dynamic states and change (often progressing) over time. We were not able to model the impact of any such change. Modelling of the impact of multimorbidity and frailty in diabetes could potentially be improved by using serial measurements, over a longer follow-up, and with measurement of additional outcomes such as retinopathy and nephropathy. Several of the baseline variables were based on self-report, however participants were supported by a study nurse in providing this information and for the multimorbidity measures we supplemented these definitions with linkage to previous hospital episodes. Finally, it is important to note that UK Biobank is not a nationally representative sample. Participants were more affluent, more likely to be White, and have fewer long-term health conditions than the national average. Our prevalence findings therefore cannot be generalized to the population as a whole, and estimates of the risk of adverse outcomes are likely to be conservative. Selection bias may also lead to collider bias, where conditioning on one criteria (UK Biobank inclusion) may bias estimates of the relationship between causally proximal variables (such as age and frailty)[32,33]. This may explain the surprising finding that the prevalence of the frailty phenotype did not rise with age as expected[26].

In conclusion, our findings demonstrate that both frailty and multimorbidity are both common and clinically important in middle-aged as well as older people with type 2 diabetes, regardless of the definition used. The greater risk of mortality, cardiovascular events, and hypoglycaemia, in people living with frailty and multimorbidity means that it is important to actively detect both frailty and multimorbidity in people with type 2 diabetes, regardless of age. However, our findings also demonstrate that guidelines for managing frailty and multimorbidity in people with type 2 diabetes may not be directly applicable to younger people, in whom the absolute mortality risk remained low even among the most frail groups. While this work further supports the idea of embedding screening for both multimorbidity and frailty as part of routine diabetes reviews, it also reinforces the need to tailor risk stratification to individual patients. This should take account of patients' age, measure used

to assess frailty or multimorbidity, and other risk factors, rather than adopting prescriptive targets and recommendations to everyone who might meet some criteria for frailty or multimorbidity.

**Reporting summary**. Further information on research design is available in the Nature Research Reporting Summary linked to this article.

## Data availability
The UK Biobank data that support the findings of this study are available from the UK Biobank (www.ukbiobank.ac.uk), subject to approval by UK Biobank. Source data for the main figures in the manuscript can be accessed as Supplementary Data 7.

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

## Acknowledgements
PH is funded by a Medical Research Council Clinical Research Training Fellowship (Grant reference MR/S021949/1).

## Author contributions
PH, JL, DM, and FSM designed the study and wrote the analysis plan. BN is the data holder under UK Biobank project 14151. PH performed the analysis. PH, BDJ, EB, JL, DM, and FSM interpreted the findings. PH wrote the first draft. PH, BDJ, EB, JL, DM, and FSMreviewed this and subsequent drafts and approved the final version for submission. PH had full access to the data. FSM is the guarantor.

## Competing interests
The authors declare no competing interests.
