## [Peer Review File · Communications Medicine]

This manuscript has been previously reviewed at another Nature Research journal. This document only contains reviewer comments and rebuttal letters for versions considered at Communications Medicine.

REVIEWERS' COMMENTS:

Reviewer #1 (Remarks to the Author):

Almost all of Reviewer 1's questions and comments have been adequately answered by the authors. There is only one question that remains unanswered concerning comparisons using concordance analysis (Kappa). Concerning these comparisons between phenotype of frailty index and frailty index, a study already performed these comparisons with Kappa test but with older general population (ELSA study) . This study could be cited: "Agreement Between 35 Published Frailty Scores in the General Population"

Reviewer #2 (Remarks to the Author):

I think the authors have been very responsive to the prior review. While the results still generally are expected, this work does raise a nuanced question of how to interpret and apply frailty/multimorbidity constructs in younger populations. One possibility for the authors to consider is that while frailty and multimorbidity do indicate gradients of risk, they may not be the best lens for deciding the intensity of diabetes management. I am reminded of work in hypertension from SPRINT where frailty indices similarly graded risk of CVD and mortality in older adults (Williamson et al. JAMA. 2016 Jun 28;315(24):2673-82. doi: 10.1001/jama.2016.7050), but did not indicate differential benefit of intensive blood control. What did seem to indicate differential benefit was cognitive function (Pajewski et al. J Am Geriatr Soc. 2020 Mar;68(3):496-504. doi: 10.1111/jgs.16272). Presumably, this would fit with the author's hypothesis, as frailty in younger populations would be much less likely to be accompanied by cognitive decline.

I was not able to find in the paper some mention of the length of follow-up, but I think the need to examine this question with longer term follow-up should be mentioned, along with examining other diabetes-relevant complications (retinopathy, neuropathy, etc.). Of course, as pointed out in the prior review, a randomized trial will be the only way to understand if relaxing glycemic targets is beneficial in younger patients with T2DM and frailty/multimorbidity.

Figure 1. Would be helpful to add percentages above the bars since you're plotting absolute counts.

Response to reviewers' comments

Reviewers' comment: black

Author response: blue

REVIEWERS' COMMENTS:

Reviewer #1 (Remarks to the Author):

Almost all of Reviewer 1's questions and comments have been adequately answered by the authors. There is only one question that remains unanswered concerning comparisons using concordance analysis (Kappa). Concerning these comparisons between phenotype of frailty index and frailty index, a study already performed these comparisons with Kappa test but with older general population (ELSA study) . This study could be cited: "Agreement Between 35 Published Frailty Scores in the General Population"

Thank you for this comment. We have added the reference above as suggested.

Reviewer #2 (Remarks to the Author):

I think the authors have been very responsive to the prior review. While the results still generally are expected, this work does raise a nuanced question of how to interpret and apply frailty/multimorbidity constructs in younger populations. One possibility for the authors to consider is that while frailty and multimorbidity do indicate gradients of risk, they may not be the best lens for deciding the intensity of diabetes management. I am reminded of work in hypertension from SPRINT where frailty indices similarly graded risk of CVD and mortality in older adults (Williamson et al. JAMA. 2016 Jun 28;315(24):2673-82. doi: 10.1001/jama.2016.7050), but did not indicate differential benefit of intensive blood control. What did seem to indicate differential benefit was cognitive function (Pajewski et al. J Am Geriatr Soc. 2020 Mar;68(3):496-504. doi: 10.1111/jgs.16272). Presumably, this would fit with the author's hypothesis, as frailty in younger populations would be much less likely to be accompanied by cognitive decline.

Thank you for these comments. We agree with the reviewer and have added the following sentence:

"While frailty and multimorbidity do indicate gradients of risk, it may be that these are not the optimal tools to assess the appropriate targets for treatment in middle-aged people."

I was not able to find in the paper some mention of the length of follow-up, but I think the need to examine this question with longer term follow-up should be mentioned, along with examining other diabetes-relevant complications (retinopathy, neuropathy, etc.). Of course, as pointed out in the

prior review, a randomized trial will be the only way to understand if relaxing glycemic targets is beneficial in younger patients with T2DM and frailty/multimorbidity.

The follow-up time (median 8 years) is stated within the methods section under 'outcomes'.

We also agree with the reviewer's point and have added the following to the discussion section:

"Modelling of the impact of multimorbidity and frailty in diabetes could potential be improved by using serial measurements, over a longer follow-up, and with measurement of additional outcomes such as retinopathy and nephropathy."

Figure 1. Would be helpful to add percentages above the bars since you're plotting absolute counts.

Thank you. We have added these to figure 1 as suggested.